# Oral Capecitabine-Vinorelbine Is Associated with Longer Overall Survival When Compared to Single-Agent Capecitabine in Patients with Hormone Receptor-Positive Advanced Breast Cancer

**DOI:** 10.3390/cancers12030617

**Published:** 2020-03-06

**Authors:** Claudio Vernieri, Michele Prisciandaro, Federico Nichetti, Riccardo Lobefaro, Giorgia Peverelli, Francesca Ligorio, Emma Zattarin, Maria Silvia Cona, Pierangela Sepe, Francesca Corti, Sara Manglaviti, Marta Brambilla, Barbara Re, Antonino Belfiore, Giancarlo Pruneri, Luigi Celio, Gabriella Mariani, Giulia Valeria Bianchi, Licia Rivoltini, Giuseppe Capri, Filippo de Braud

**Affiliations:** 1Fondazione Istituto FIRC di Oncologia Molecolare (IFOM), Via Adamello 16, 20133 Milan, Italy; 2Medical Oncology Unit, Fondazione IRCCS Istituto Nazionale dei Tumori, Via Venezian 1, 20133 Milan, Italy; michele.prisciandaro@istitutotumori.mi.it (M.P.); federico.nichetti@istitutotumori.mi.it (F.N.); riccardo.lobefaro@istitutotumori.mi.it (R.L.); giorgia.peverelli@istitutotumori.mi.it (G.P.); francesca.ligorio@istitutotumori.mi.it (F.L.); emma.zattarin@istitutotumori.mi.it (E.Z.); pierangela/sepe@istitutotumori.mi.it (P.S.); francesca.corti@istitutotumori.mi.it (F.C.); sara.manglaviti@istitutotumori.mi.it (S.M.); marta.brambilla@istitutotumori.mi.it (M.B.); luigi.celio@istitutotumori.mi.it (L.C.); gabriella.mariani@istitutotumori.mi.it (G.M.); giulia.bianchi@istitutotumori.mi.it (G.V.B.); giuseppe.capri@istitutotumori.mi.it (G.C.); filippo.debraud@istitutotumori.mi.it (F.d.B.); 3ASST Fatebenefratelli Sacco—PO Luigi Sacco, Via G.B. Grassi 74, 20133 Milan, Italy; cona.silvia@asst-fbf-sacco.it; 4SC Pharmacy, Fondazione IRCCS Istituto Nazionale dei Tumori, Via Venezian 1, 20133 Milan, Italy; barbara.re@istitutotumori.mi.it; 5Department of Diagnostic Pathology and Laboratory Medicine, Fondazione IRCCS Istituto Nazionale dei Tumori, 20133 Milan, Italy; antonino.belfiore@istitutotumori.mi.it (A.B.); giancarlo.pruneri@istitutotumori.mi.it (G.P.); 6Department of Oncology and Hemato-Oncology, University of Milan, 20133 Milan, Italy; 7Unit of Immunotherapy of Human Tumors, Fondazione IRCCS Istituto Nazionale dei Tumori, 20133 Milan, Italy; licia.rivoltini@istitutotumori.mi.it

**Keywords:** advanced breast cancer, chemotherapy, capecitabine, vinorelbine, hormone receptor-positive breast cancer, triple-negative breast cancer, progression-free survival, overall survival, adverse events

## Abstract

*Background*: Single-agent capecitabine (C) is a moderately effective chemotherapeutic compound in the treatment of patients with HER2-negative metastatic breast cancer (mBC). The capecitabine-vinorelbine (CV) combination is also used due to a good tolerability profile, but no studies have demonstrated its superiority over single-agent C. *Methods*: We conducted a retrospective analysis to compare overall response rate (ORR), progression-free survival (PFS), overall survival (OS) and incidence of adverse events (AEs) in patients with HER2-negative mBC treated with CV vs. single-agent C. *Results*: Out of 290 patients included in this study, 127 (43.8%) received single-agent C, while 163 (56.2%) patients were treated with CV. Median PFS was similar in patients treated with single-agent C or CV, while CV was associated with significantly longer OS in patients with hormone receptor-positive (HR+) BC. This OS advantage was confirmed at multivariable analysis also after propensity score-based matching of patients according to relevant clinical or tumor characteristics. When compared with single-agent C, CV was associated with higher incidence of G3/G4 and any-grade nausea/vomiting, diarrhea and increased transaminases. *Conclusions*: While prospective studies are needed to confirm our findings, the potential OS advantage of CV over single-agent C in HR+ mBC patients must be weighed against a significantly higher incidence of AEs.

## 1. Introduction

In recent years, the therapeutic armamentarium against HER2-negative (HER2−) metastatic breast cancer (mBC) has remarkably expanded [1]. In particular, the introduction of Cyclin-Dependent Kinase 4/6 (CDK 4/6) inhibitors and PIK3CA/mTORC1 inhibitors in combination with endocrine treatments contributed to significantly prolong disease control in patients with advanced hormone receptor-positive (HR+) BC, while the PD-L1 inhibitor atezolizumab in combination with nab-paclitaxel recently improved the progression-free survival (PFS) of patients with advanced triple-negative breast cancer (TNBC) in the first-line setting [2,3,4,5,6,7].

Despite these progresses, cytotoxic chemotherapy (ChT) remains a mainstay for the treatment of almost all patients with HER2-negative mBC: in TNBC, single-agent or combination ChT are the only available therapeutic options for the vast majority of patients in all treatment lines; in HR+ HER2- BC with primary or acquired resistance to endocrine therapies, ChT is the only effective treatment to control disease growth, to relieve patient symptoms and to prolong survival [1].

Since the vast majority of mBC patients have received anthracycline- and/or taxane-based ChT in the (neo)adjuvant therapy setting, or as previous lines for advanced disease treatment, other classes of cytotoxic agents are often needed to avoid cross-resistance and cumulative toxicities [8,9,10]. Among them, the fluoropyrimidine capecitabine (C) [11,12] and the vinca alkaloid vinorelbine (V) [13] combine good safety profiles with acceptable anticancer activity even in heavily pre-treated patients. In addition, the oral formulation of C and V avoids the need for intravenous administration and results in improved patient comfort and lower hospital accesses.

In mBC, single-agent ChT is usually the preferred treatment option. Indeed, although combination ChT has been associated with higher rates of tumor responses and PFS prolongation in some clinical studies, it is burdened by a higher incidence of adverse events (AEs) and its impact on overall survival remains uncertain [14,15,16]. The oral capecitabine-vinorelbine (CV) combination may represent an exception in this context: indeed, due to its acceptable toxicity profile and antitumor activity, it is frequently used in the clinical practice in mBC patients previously treated with anthracyclines plus/minus taxanes [17,18,19,20]. One recent pooled analysis of prospective phase II/III trials showed that CV is safe and effective, with a median PFS of 7.3 and 3.8 months in the first- and second-line treatment settings, respectively [21]. However, no studies have directly compared CV with single-agent C, which remains one of the most effective therapeutic options, in terms of both efficacy and toxicity, in mBC patients. Therefore, it is unclear if CV may be superior to C in patients with specific mBC subtypes.

In this study, we conducted a monocentric, retrospective analysis to compare the antitumor activity (overall response rate or ORR; disease control rate or DCR), efficacy (PFS, OS), antitumor activity and safety profile of CV vs. single-agent C in a population of 290 consecutive HER2− mBC patients.

## 2. Results

### 2.1. Patient Population

Main patient and tumor characteristics are described in Table 1. Out of 290 patients included in this study, 127 (43.8%) received C and 163 (56.2%) received the CV combination.

Patients treated with CV were more likely to be younger (*p* = 0.01) and to have received previous anthracycline- (*p* = 0.01) and/or taxane-based (*p* = 0.01) ChT for limited-stage and/or advanced disease. Treatment subgroups were well balanced in terms of Eastern Cooperative Oncology Group (ECOG) performance status (PS) (0–1 vs. 2), tumor biology (HR+ BC vs. TNBC), line of ChT for advanced disease (1 vs. >1), number of metastatic sites (1 vs. >1), presence of visceral metastases (yes vs. no), synchronous/metachronous metastases at the diagnosis of metastatic disease, maintenance endocrine treatment (yes vs. no). CV and C were administered as the first-line treatment for advanced disease in 46.6% and 48% of patients, respectively. There was no significant association between tumor biology (TNBC vs. HR+ BC) and the number of previous chemotherapy lines (χ^2^ test *p* value = 0.68) or previous exposure to anthracycline- (χ^2^ test *p* value = 0.72); on the other hand, TNBC patients were more likely to have received previous taxanes based chemotherapy when compared to HR+ mBC patients (χ^2^ test *p* value = 0.0036).

### 2.2. Treatment Activity and Efficacy

Median follow-up of patients included in this study was 80.7 months (95% CIs 68.1–94.8 months). At the time of data cut-off and analysis, in the whole patient population the ORR and DCR were 33.8% and 59%, respectively. In the subgroup of HR+ BC patients, ORR and DCR were 35.7% and 64.3%, respectively. We found no statistically significantly different ORR or DCR between patients treated with CV vs. C (ORR: 36.8% vs. 29.9%, *p* = 0.269; DCR: 60.7% vs. 56.7%, *p* = 0.566, respectively). In the subgroup of HR+ BC patients, there was a trend towards higher ORR with CV vs. C (43.3% vs. 28.7%; *p* = 0.071), but no differences in terms of DCR (68.2% vs. 59.4%; *p* = 0.21).

With a total number of 279 progression events, median PFS was 6.37 months (95% CIs: 5.55–7.39 months) in the whole patient population, with a trend towards longer PFS in patients with HR+ BC when compared to patients with TNBC (PFS: 7.02 vs. 3.52 months, *p* = 0.088) (Appendix A). We found no significant PFS differences between patients treated with CV or C (median PFS: 6.48 and 6.12 months, respectively; *p* = 0.7) (Figure 1A). However, there was a trend towards better PFS with CV over single-agent C in the subgroup of patients with HR+ BC (median PFS: 7.56 vs. 6.48 months, respectively; *p* = 0.15) (Figure 1B), but not in patients with TNBC (Appendix A).

With a total number of 236 death events, median OS was 23.9 months (95% CIs 21–27 months) in the whole patient population, and was significantly longer in HR+ BC patients than in TNBC patients (OS: 25.9 vs. 12.3 months; *p* = 0.005) (Appendix AC). Overall, there was a trend towards better OS in patients treated with CV when compared to patients receiving single-agent C (median OS: 26.4 vs. 20.4 months, respectively; *p* = 0.09) (Figure 1C), which reached statistical significance in the subgroup of patients with HR+ BC (median OS: 27.6 vs. 22.7 months, respectively; *p* = 0.02) (Figure 1D), but not in patient with TNBC (Appendix AD).

Among HR+ BC patients, those with estrogen receptor (ER)-positive and progesterone receptor (PgR)-positive (ER+ PgR+) disease had significantly longer PFS and OS when treated with CV vs. C (median PFS: 7.28 vs. 5.96 months, *p* = 0.025; median OS: 27.2 vs. 20.9 months, *p* = 0.01) (Appendix A), while we found no significant PFS or OS differences among ER+ and PgR-negative (PgR-) BC patients treated with CV vs. C (median PFS: 9.63 vs. 13.08 months, *p* = 0.53; median OS: 32 vs. 27.7 months, *p* = 0.87) (Appendix A). There were no patients with ER-negative (ER-) and PgR+ BC in this study.

### 2.3. Factors Independently Associated with Patient Prognosis

The following factors were associated with significantly higher risk of disease progression at univariate analysis: TNBC biology (HR 1.3), the presence of visceral metastases (HR 1.27), worse ECOG PS (HR 2.72), a higher number of metastatic sites (HR 1.44), the presence of synchronous metastases at diagnosis (HR 1.44) (Figure 2A). Multivariable analysis adjusting for these factors confirmed an independent association between worse PFS and the following factors: TNBC biology (HR 1.46), worse ECOG PS (HR 2.54), a higher number of metastatic sites (HR 1.49), the presence of synchronous metastases at diagnosis (HR 1.45) (Figure 2B).

Predictors of worse OS at univariate analysis were: TNBC biology (HR 1.56), the presence of visceral disease (HR 1.44), single-agent ChT (i.e., C vs. CV: HR 1.25), worse ECOG PS (HR 2.39), a higher number of metastatic sites (HR 1.75), the presence synchronous metastases at diagnosis (HR 1.37) (Figure 2C). At multivariable analysis, TNBC biology (HR 1.66), single-agent C (HR 1.37), worse ECOG PS (HR 2.14) and a higher number of metastatic sites (HR 1.73) confirmed their independent association with shorter OS (Figure 2D).

To evaluate the impact of ChT on OS in more homogeneous patient populations, we performed multivariable analysis in a subgroup of 238 patients (119 treated with CV and 119 with C) matched for variables that were independently associated with OS, i.e., TNBC biology, ECOG PS and number of metastatic sites (Criterion a, see Material and Methods). In this population (see Appendix A for clinical/tumor characteristics), covariates independently associated with higher risk of death were: C treatment (HR 1.36), TNBC biology (HR 1.35) and a higher number of metastatic sites (HR 1.96) (Figure 3A). Furthermore, with an HR of 1.63 and 1.3, respectively, single-agent C treatment and TNBC biology were the only factors associated with significantly worse OS in 176 (88 treated with CV and 88 with C) patients matched on the basis of covariates that were unequally distributed in the CV vs. C subgroups (i.e., age, previous anthracycline and taxane treatment: Appendix A; Figure 3B).

### 2.4. Impact of Study Treatments on the Neutrophil-to-Lymphocyte Ratio (NLR)

Trying to explain the observed OS advantage of CV treatment in HR+ BC patients, we hypothesized that CV might impact on the immune system ability to control tumor growth in the long-term period. If correct, this hypothesis could account for the OS benefit associated with CV versus C, even in the absence of PFS benefit. To test this hypothesis, we investigated the impact of CV and C on the neutrophil-to-lymphocyte ratio (NLR), a biomarker that has been consistently associated with worse clinical outcome in patients with both limited-stage and advanced BC [22]. In particular, we hypothesized that CV reduces the neutrophil-to-lymphocyte ratio (NLR) significantly more than single-agent C. By using the Maximally Selected Rank Statistics, we identified 3.67 as the best NLR threshold associated with the most significant separation of Kaplan Meier OS curves in the population of enrolled patients (Appendix A). Patients with NLR higher than 3.67 at baseline had significantly worse OS at multivariable analysis, thus confirming that high NLR is an independent prognostic factor in mBC patients (Appendix A). At baseline, patients treated with CV or C had non-significantly different NLR (median 2.8 vs. 2.93 *p* = 0.25). Of note, after the first ChT cycle the NLR was significantly lower in patients treated with CV than in patients treated with single-agent C (median 1.86 vs. 2.43; *p* < 0.0001). These data demonstrate that CV significantly and precociously down-modulates a biomarker associated with worse survival in mBC patients.

### 2.5. Treatment Safety and Tolerability

Treatment-related AEs are summarized in Table 2. Any-grade AEs occurred in 85.9% of the whole patient population, without significant differences between patients treated with C (83.5%) or CV (89%). The most common AEs consisted in: any grade neutropenia (42.1%), any grade anemia (44.9%), increased liver transaminases (33.4%), nausea/vomiting (27.2%), diarrhea (22.1%), hand-foot syndrome (21%). The following AEs were significantly more common in patients treated with CV than in patients receiving single-agent C: any-grade neutropenia (54.6% vs. 26%; *p* = 0.014); G3/G4 neutropenia (14.1% vs. 0.8%; *p* = 0.018); nausea/vomiting (31.3% vs. 22%; *p* = 0.001); diarrhea (24.5% vs. 18.9%; *p* = 0.046) and increased blood levels of transaminases (35.6% vs. 29.9%; *p* = 0.041). Other AEs were similar in the two treatment groups.

## 3. Discussion

C is a well-tolerated, manageable and effective ChT agent in the treatment of HER2- mBC patients. The CV combination is also frequently used in the clinical setting due to good manageability and safety [21], but it has been never compared with single-agent C in terms of antitumor efficacy. Here, we conducted the first retrospective, monocentric study to compare the efficacy and safety profiles of combination CV vs. single-agent C.

We did not find significant PFS differences between patients treated with CV or C. Although in the subgroup of HR+ BC patients there was a trend towards longer PFS in the combination treatment cohort, multivariable analysis did not confirm an independent role of CV treatment on PFS. Regarding OS, CV was associated with significantly lower risk of death independently of other relevant clinical or tumor-related variables, with the survival advantage being limited to patients with HR+ BC, and in particular to patients with ER+ PgR+ disease. Of note, median PFS (6.48 months) and OS (26.4 months) in the CV cohort of our study were in line with data of previous prospective phase II trials [17,18,19,20], as well as with data of a recently published study, in which the CV combination confirmed to be effective also in patients with heavily pretreated mBC [23].

To explain the observed OS advantage in CV-treated vs. C-treated patients despite the lack of PFS benefit, we hypothesized that patients receiving the CV combination might have more favorable clinical/tumor characteristics. Therefore, we performed a propensity score analysis to match treatment assignment (CV vs. C) with covariates that were imbalanced in the two treatment groups, or that were independently associated with patient OS. Of note, multivariable analysis conducted after propensity score-based matching of patients confirmed an independent association between CV and longer OS. Therefore, CV could provide a true survival benefit in HR+ BC patients despite the lack of PFS superiority. Larger prospective studies are required to confirm this interesting observation in patients with advanced HR+ mBC.

In a recently published monocentric study comparing combination carboplatin-gemcitabine (CG) with single-agent gemcitabine (G) in mBC patients, we found that CG was associated with longer OS despite the lack of significant PFS differences [24]. However, the high clinical heterogeneity between CG- and G-treated patients, and in particular the fact that patients receiving CG were less heavily pre-treated, provides a reasonable and simplistic explanation to results of our prior analysis. In the current study, treatment groups were quite well balanced in terms of clinical/tumor characteristics, the number of enrolled patients was higher, and multivariable analysis in the whole patient population, as well as in subgroups of patients matched on the basis of propensity scores, confirmed an independent association between CV and longer patients OS. These arguments reinforce our conclusion that CV may provide a true OS benefit when compared to single-agent C.

In our study, the separation of OS Kaplan Meier curves of HR+ BC patients treated with CV or C occurred quite precociously (i.e., 1–3 months) after therapy initiation, and even increased during the treatment course (Figure 1D). On the other hand, Kaplan Meier PFS curves of CV- and C-treated patients overlapped during the first months of treatments, with only a non-significant separation during the subsequent treatment months. For these reasons, we tend to exclude that death events in the CV cohort were delayed as a result of delayed disease progression events. Rather, the CV combination may induce changes in tumor biology or immune system activation that could condition a less aggressive disease course and/or better response to treatments administered during subsequent lines.

In two prospective, randomized phase III trials, the non-taxane microtubule inhibitor eribulin mesylate was associated with significantly longer OS, or with a trend towards longer OS, when compared to C or other treatments of physician choice, despite the lack of significant benefit in terms of PFS [12,25]. Potential mechanisms to explain these clinical data consist in eribulin’s ability to revert epithelial-to-mesenchymal transition (EMT) and metastatic spread of breast cancer cells [26], to enhance immune system activation [27] or to remodel tumor vasculature [28,29]. Since there are no published preclinical data supporting these mechanistic explanations in the case of CV, in this study we preliminarily explored the hypothesis that CV reshapes systemic immune system activation. To do so, we evaluated the precocious effects of CV and C on the NLR, a biomarker that recapitulates systemic inflammation and immune system activation and is associated with worse prognosis in mBC patients [22,30]. Consistent with data from the literature, we found that higher NLR at baseline correlated with worse OS. Of note, patients receiving CV had significantly lower NLR after one treatment cycle when compared to patients treated with C. These findings indicate a potential immunoregulatory effect of the CV combination and suggest that CV-induced immune system modulation may at least in part explain the observed association with longer patient survival. Prospective studies are needed to confirm and consolidate these data, as well as to explore the effect of C and CV on specific subpopulations of myeloid-derived suppressor cells (MDSCs) that are associated with worse mBC patient prognosis [31,32], or on antitumor lymphocytic effectors associated with higher treatment efficacy [33].

Since TNBC is typically more aggressive and rapidly growing when compared with HR+ BC, we expected a higher clinical benefit with CV over single-agent C in this patient cohort. Therefore, we were surprised that the OS advantage associated with CV was limited to HR+ BC patients. Reasonable hypotheses to explain these findings include: (1) higher sensitivity of TNBC to fluoropyrimidines, including C, which could make the anticancer effect of concomitant V negligible [34]; (2) limited number of TNBC patients enrolled, with low power to detect OS differences between patients treated with CV or C; (3) the fact that HR+ mBC patients were less likely to have received prior taxane-based ChT in the (neo)adjuvant or advanced disease treatment settings when compared to TNBC patients. Since taxanes and V act by disrupting microtubule dynamics (although with different mechanisms of action), lower prior exposure to taxanes therapy could be associated with higher benefit from CV; (4) different biological characteristics between HR+ BC and TNBC, which could make HR+ BC more sensitive to the CV combination. In this regard, it is important to note that C and V have very different antitumor mechanisms of action. C is the precursor of the antimetabolite 5-fluorouracil (5-FU), which inhibits the thymidylate synthase enzyme and DNA synthesis, thus preventing cell cycle progression through the S phase [35]. On the other hand, V is a spindle poison that binds β-tubulin and inhibits microtubule polymerization, thus resulting in the activation of the Spindle Assembly Checkpoint (SAC) and in the inhibition of cell cycle progression through the G2/M phase [36,37]. Therefore, C and V have potentially synergistic antitumor activity deriving from their action in two different phases of the cell cycle. HR+ BC is less sensitive to chemotherapy when compared to TNBC, likely due to lower growth and proliferation rates, which result in slower progression through different phases of the cell cycle [38]. Combining two cytotoxic compounds that act in different phases of the cell cycle, such as C and V, could significantly increase the chances to halt cell cycle progression in the relatively chemo-resistant and slow-proliferating HR+ BC subtype. To reinforce this hypothesis are results of our subgroup analysis conducted in HR+ BC patients, which revealed a significant advantage, in terms of both PFS and OS, with CV over C specifically in ER+ PgR+, but not in ER+ PgR- patients (Appendix A). This finding indicates that luminal A-like (ER+ PgR+) BCs, which are characterized by slower tumor cell growth and proliferation rates, could be more sensitive to the CV combination when compared with luminal B-like (ER+ PgR−) BCs. In our opinion, this hypothesis deserves further investigation in preclinical studies and, in case, in prospective clinical studies in patients with HR+ BC.

Regarding treatment-related AEs, the CV combination was associated with significantly higher incidence of any grade and G3/G4 neutropenia, any grade nausea/vomiting, diarrhea and increase in liver transaminases when compared to single-agent C. Other AEs were not significantly different in the CV and C cohorts. Of note, the observed incidence of G3/G4 neutropenia in the CV cohort is in line with results of a recent pooled analysis of prospective trials [21], and is consistent with lower NLR values after one treatment cycle.

Strengths of this study consist in: (a) its monocentric design, which guarantees homogeneous patient treatment and follow-up; (b) the relatively high number of patients evaluated in a 7-year time interval; (c) globally low patient heterogeneity in CV or C cohorts; (d) the fact that OS results were confirmed at multivariable analysis in subgroups of patients matched according to their propensity score of covariate distributions in CV and C cohorts. Main study limitations consist in the retrospective nature of the study.

## 4. Materials and Methods

### 4.1. Study Setting and Inclusion Criteria

This was a monocentric, retrospective study conducted in patients with HER2- mBC treated between January 2012 and December 2018 at Fondazione IRCCS Istituto Nazionale dei Tumori (Milan, Italy) with single-agent C or the CV combination. Subjects were considered eligible if all the following criteria were fulfilled: (1) women with pathologically/cytologically confirmed diagnosis of unresectable locally recurrent or metastatic HER2− BC; (2) age ≥ 18 years; (3) ECOG PS of 0–2; (4) treatment with one of the following ChT regimens: oral C at the dosage of 1000 mg/m^2^ twice per day on days 1 to 14 in 21-day cycles (single-agent), or C at the dosage of 1000 mg/m^2^ twice per day on days 1 to 14 in combination plus oral V at the dosage of 60 mg/m^2^ on days 1 and 8 in 21-day cycles (CV regimen); (5) available information about previous therapies for both limited-stage and/or advanced disease; (6) available data regarding patient outcomes with the study treatments, including objective response rate (ORR), PFS and OS; (7) availability of medical records for the collection of data regarding treatment-related adverse events (AEs).

Among HER2− BC patients included in this study, we also evaluated the antitumor activity/efficacy of CV and C in the subgroups of patients with triple-negative breast cancer (TNBC) and hormone receptor-positive breast cancer (HR+ BC). TNBC was defined on the basis of an expression of estrogen receptor (ER) and progesterone receptor (PgR) in less than 1% of cancer cells, while HR+ BC was defined by the presence of ER and/or PgR in at least 1% of evaluated tumor cells. Among HR+ BC patients, we also performed a sub-analysis to assess the antitumor activity/efficacy of CV vs. C in subgroups of patients with ER-positive and PgR-positive (ER+ PgR+), as well as in patients with ER+ and PgR-negative (PgR−) disease.

### 4.2. Objectives of the Study

The objective of the study was to compare the antitumor activity/efficacy and safety profiles of oral CV vs. single-agent C in patients with HER2− mBC in the “real life” setting. The primary study endpoint was PFS, as defined as the time between treatment initiation and disease progression or patient death from any cause, whichever came first. Some HR+ BC patients received maintenance endocrine therapy after obtaining the best response with C or CV, or due to ChT-related toxicities. In these cases, PFS was defined as the time between ChT initiation and disease progression or patient death during the administration of maintenance therapy or after its discontinuation for any reason. Overall response rate (ORR), disease control rate (DCR) and OS were secondary efficacy endpoints of this study. ORR was defined as the proportion of patients achieving partial response (PR) or complete response (CR) as their best response. DCR was defined as the proportion of patients achieving PR, CR or stable disease (SD) as their best response. OS and was defined as the time between initiation of the study treatment and death from any cause.

ChT was administered until disease progression, unacceptable toxicity or patient decision. After a variable number of CV cycles, some patients in the combination cohort continued their treatment with single-agent C or V after obtaining the best response, or as a result of ChT-related AEs. Since these patients continued to receive part of the initial therapy, they were not considered as patients receiving “maintenance” treatment.

The study protocol was approved by the Internal Review Board (IRB) and the Local Ethics Committee of Fondazione IRCCS Istituto Nazionale dei Tumori (study registration code: INT 182/19). Patient data were collected according to the ethical principles for medical research involving human subjects adopted in the Declaration of Helsinki. Patients alive at the time of data collection and/or analysis signed an informed consent form, which was approved by the IRB, for the use of their personal data for research purposes. The IRB also authorized the collection and analysis of data from patients who were not alive at the time of data collection and analysis; in these cases, no additional consent forms were required by the IRB for the use of these data, except for the general consent form that all patients sign when starting a treatment at our Institution, in which they authorize the use of data collected as per clinical practice for research purposes.

### 4.3. Assessment of Efficacy and Safety

Tumor response was assessed every three ChT cycles through computed tomography (CT) or magnetic resonance imaging (MRI), but tumor re-evaluation was anticipated if evolving symptoms or other clinical signs indicative of progressive disease (PD) emerged. We used the Response Evaluation Criteria in Solid Tumors (RECIST, version 1.1) to assess tumor response. Clinically evident lesions were evaluated by physical examination, with every three-week measurement of lesion diameters using calipers.

To assess treatment safety, we recorded all AEs from medical records and blood exams. AEs were classified according to the common terminology criteria for AEs (CTCAE), Version 5.0 of November 2017, National Institutes of Health, National Cancer Institute. Hematologic toxicities were reviewed at the moment of data collection from computerized blood sample data. Non-hematologic toxicities were extracted from medical records, where they are regularly annotated during patient visits. Since the grading of patient symptoms in retrospective studies may be not fully reliable, mainly due to the AE under-reporting or under-grading, in this study non-hematologic AEs were reported as any-grade AEs, as previously described [24].

### 4.4. Exploratory Analyses

Given the retrospective nature of this study, patients were not randomly assigned to receive CV or C, with the risk of imbalanced distribution of crucial clinical/tumor parameters in the two treatment cohorts. To minimize the potential impact of these imbalances on the study results, we used the propensity score methodology. In particular, we adjusted the association between the type of treatment (CV vs. C) and OS for the likelihood that a patient had of receiving that treatment, given specific baseline characteristics (covariates) [1,39]. We chose two different sets of covariates: (a) covariates independently associated with patients OS in the general multivariable model (i.e., TNBC biology; ECOG PS; number of metastatic sites); (b) covariates unequally distributed in CV vs. C cohorts (i.e., age; previous anthracycline therapy; previous taxane therapy).

To assess the impact of parameters associated with systemic inflammation/immune system activation on the efficacy of different treatments, we tested the impact of the Neutrophil-to-Lymphocyte Ratio (NLR) on patient OS. The Maximally Selected Rank Statistics was used to find the best threshold to discriminate patients with high vs. low NLR. The NLR was calculated at baseline and after one ChT cycle, and median values were compared in patients treated with CV or C.

### 4.5. Statistical Analyses

We used the χ^2^ test was used to study the distribution of individual dichotomous variables (patient- or tumor-related) in the CV vs. C treatment cohorts, as well as to investigate the association between tumor biology (TNBC vs. HR+ BC) and previous treatment exposure (i.e., number of lines of treatment, previous exposure to anthracyclines or taxanes). We used the proportion test to compare the ORR and the rate of AEs in the two cohorts. The Wilcoxon test was used to compare median NLR values at baseline or one month after treatment initiation in patients treated with C or CV. PFS and OS were represented according to the Kaplan–Meier method, and survival distributions were compared with the log-rank test. Patients who had not undergone disease progression or death at the time of data cut-off and analysis were censored at their last disease evaluation. The impact of different prognostic factors on PFS and OS was first assessed at univariate analysis. Factors significantly associated with the risk of progression or death were included in a Cox proportional hazard model to assess their independent association with survival.

For propensity score-based patient matching, we calculated a propensity score for each patient and for each of the two sets of covariates (see “Exploratory analyses”). Only patients with similar propensity score values were included in the matched groups, with a 1:1 matching ratio between the two study cohorts (CV and C) and a 0.1 caliper (i.e., the maximum accepted difference in the propensity scores of matched subjects).

A threshold of 0.05 was set a significance threshold for all statistical analyses, with the exception of univariate analyses, for which we used a threshold of 0.1. We used a higher threshold for determining statistical significance at univariate analysis because we wanted to be less stringent in the selection of covariates that were then tested in the multivariable Cox model. All statistical analyses were performed using the software R (version 3.3.2 (2016-10-31)) [24].

### 4.6. Data Availability

The datasets generated and/or analysed during the current study are available from the corresponding author on reasonable request.

## 5. Conclusions

While confirming previous data of safety and efficacy of C and CV in HER2- BC patients, this study showed for the first time that CV is associated with significantly longer OS when compared to single-agent C specifically in the subgroup of HR+ mBC patients. Since this OS advantage was not paralleled by a similar advantage in terms of PFS, single-agent C should remain a preferred treatment option for the vast majority of anthracycline- and taxane-pretreated mBC patients, especially in more advanced lines of treatment. However, the observed OS benefit and acceptable safety profile of CV make this combination a valid treatment option for fit patients with HR+ mBC. Preclinical and clinical studies are needed to explore the hypothesis that the CV combination may impact on antitumor immunity by reducing the number and activation status of peripheral blood and intratumor MSDCs and immunosuppressive macrophages, while increasing antitumor effectors.

## Figures and Tables

**Figure 1 cancers-12-00617-f001:**
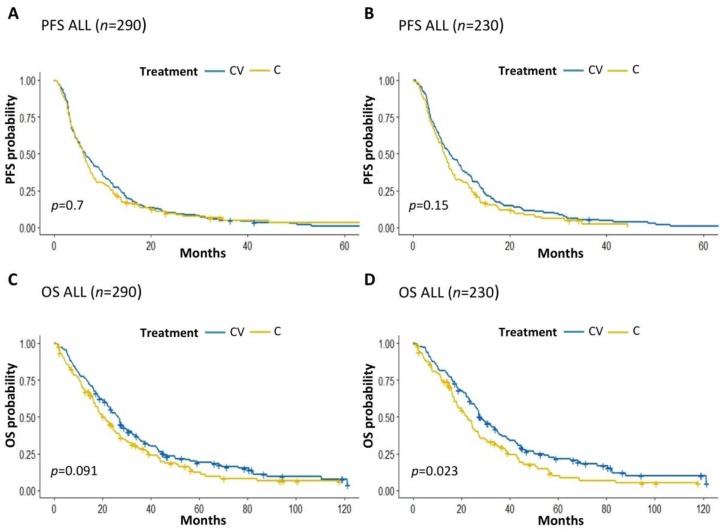
(**A**,**B**): Kaplan Meier curves of PFS in the whole population of enrolled patients treated with CV vs. C (**A**) or in HR+ BC patients treated with CV vs. C (**B**). (**C**,**D**): Kaplan Meier curves of OS in the whole population of enrolled patients treated with CV vs. C (**C**) or in the subgroup of HR+ BC patients treated with CV vs. C (**D**). C: capecitabine; V: vinorelbine; PFS: Progression-free survival; OS: Overall survival. The + symbol indicates patients censored at the time of data cut off and analysis.

**Figure 2 cancers-12-00617-f002:**
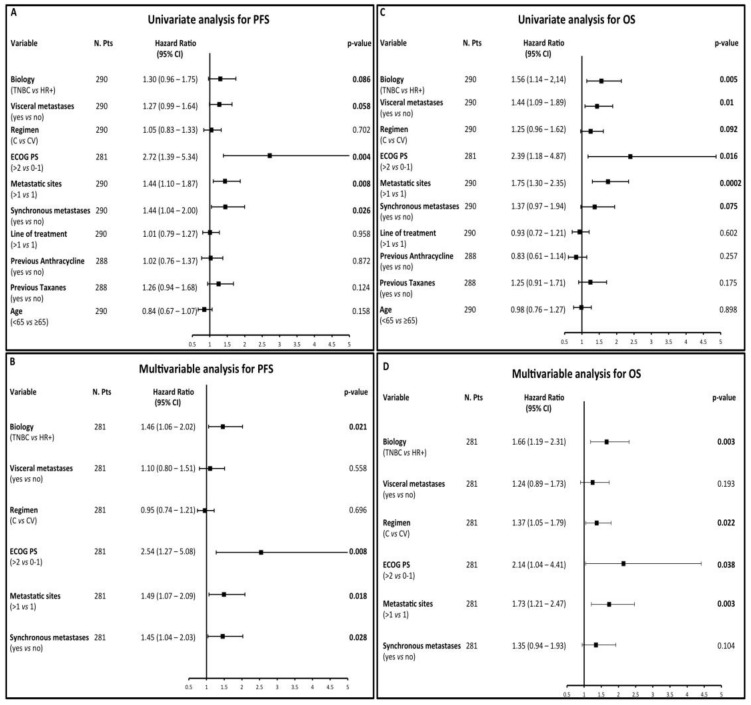
(**A**–**D**) Forest plots summarizing the results of univariate (**A**) and multivariable (**B**) analysis of patient progression-free survival (PFS) according to the treatment received and other meaningful covariates, as well as of univariate (**C**) and multivariable (**D**) analysis of patient overall survival (OS) according to the treatment received and other relevant covariates. C: capecitabine; CV: capecitabine-vinorelbine.

**Figure 3 cancers-12-00617-f003:**
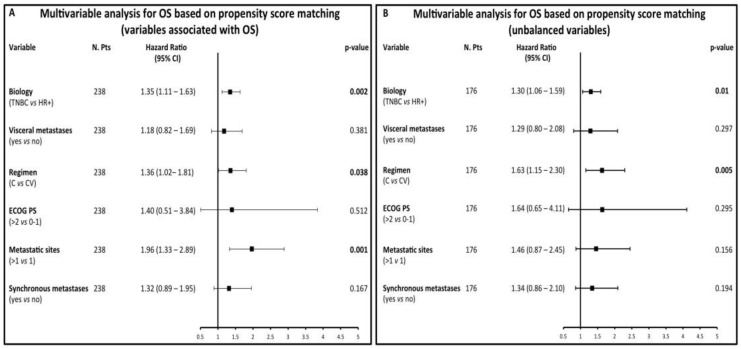
(**A**,**B**): multivariable analysis investigating the independent association between the type of treatment received (CV vs C) and other meaningful variables in patient populations matched on the basis of their propensity scores according to clinical/tumor characteristics independently associated with OS (**A**) or that are imbalanced in the CV and C treatment cohorts (**B**).

**Table 1 cancers-12-00617-t001:** Clinical and tumor characteristics in the overall population and in the C and CV subgroups.

Characteristic	Overall	C Subgroup	CV Subgroup	*p*
	*n* = 290	*n* = 127	*n* = 163	
**Age, Years**				**<0.001**
**≥65**	134 (46.2)	85 (66.9)	49 (30.1)	
**<65**	156 (53.8)	42 (33.1)	114 (69.9)	
**ECOG PS**				
**0–1**	281 (96.9)	120 (94.5)	161 (98.8)	
**≥2**	9 (3.1)	7 (5.5)	2 (1.2)	0.08
**Tumor Biology**				
**HR+**	230 (79.3)	101 (79.5)	129 (79.1)	
**TNBC**	60 (20.7)	26 (20.5)	34 (20.9)	1.00
**Line of ChT for mBC**				
**1**	137 (47.2)	61 (48.0)	76 (46.6)	
**>1**	153 (52.8)	66 (52.0)	87 (53.4)	0.91
**N. of metastatic sites**				
**1**	80 (27.6)	38 (29.9)	42 (25.8)	
**>1**	210 (72.4)	89 (70.1)	121 (74.2)	0.51
**Visceral disease**				
**Yes**	192 (66.2)	83 (65.4)	109 (66.9)	
**No**	98 (33.8)	44 (34.6)	54 (33.1)	0.88
**Time to Metastases**				
**Synchronous**	44 (15.2)	19 (15.0)	25 (15.3)	
**Metachronous**	246 (84.8)	108 (85.0)	138 (84.7)	1.00
**Previous Anthracycline** **Treatment**				
**Yes**	228 (78.6)	88 (69.3)	140 (85.9)	
**No**	62 (21.4)	39 (30.7)	23 (14.1)	**0.001**
**Previous Taxane Treatment**				
**Yes**	227 (78.2)	88 (69.3)	139 (85.3)	
**No**	63 (21.8)	39 (30.7)	24 (14.7)	**0.002**
**Maintenance Treatment**				
**Yes**	47 (16.2)	18 (14.2)	29 (17.8)	
**No**	243 (83.8)	109 (85.8)	134 (82.2)	0.50
**C/V in subsequent lines**				
**Yes**	83 (28.6)	42 (33.1)	41 (25.2)	
**No**	207 (71.4)	85 (66.9)	122 (74.8)	0.18

Data are presented as *n* (%) except where otherwise noted. The *p* value of the χ^2^ test assessing the association between each characteristic and the type of treatment received is indicated in the right column of the table. The *p* value of the test is indicated in bold numbers when statistically significant. Abbreviations: C: capecitabine; CV: capecitabine-vinorelbine combination; ECOG PS: Eastern Cooperative Oncology Group Performance Status; HR+ BC: hormone receptor-positive breast cancer; TNBC: triple-negative breast cancer. Bold: statistically significant difference.

**Table 2 cancers-12-00617-t002:** Rate of AEs that occurred in patients treated with capecitabine or capecitabine-vinorelbine.

AEs	Overall	C Subgroup	CV Subgroup	*p*
	*n* = 290	*n* = 127	*n* = 163	
**Any**	249 (85.9)	106 (83.5)	145 (89.0)	
**Neutropenia**				
**Any Grade**	122 (42.1)	33 (26.0)	89 (54.6)	**0.014**
**Grade 3–4**	24 (8.3)	1 (0.8)	23 (14.1)	**0.018**
**Lymphopenia**	77 (26.6)	31 (24.4)	46 (28.2)	0.087
**Anemia**				
**Any Grade**	135 (46.6)	57 (44.9)	78 (47.9)	0.070
**Grade 3–4**	1 (0.3)	1 (0.8)	\	\
**Thrombocytopenia**				
**Any Grade**	42 (14.5)	20 (15.7)	22 (13.5)	0.757
**Grade 3–4**	2 (0.7)	1 (0.8)	\	\
**Nausea/Vomiting**	79 (27.2)	28 (22.0)	51 (31.3)	**0.001**
**Constipation**	13 (4.5)	6 (4.7)	7 (4.3)	0.782
**Diarrhea**	64 (22.1)	24 (18.9)	40 (24.5)	**0.046**
**Hand-foot skin reaction**	61 (21.0)	28 (22.0)	33 (20.2)	0.522
**Increased liver transaminases**	97 (33.4)	38 (29.9)	58 (35.6)	**0.041**
**Other toxicities**	90 (31.0)	56 (44.1)	62 (38.0)	0.581

Data are presented as *n* (%) except where otherwise noted. The *p* value of the χ^2^ test assessing the association between each AE and the type of treatment received is indicated in the right column of the table. *p* values are shown in bold when statistically significant. Abbreviations: AE: adverse event; C: capecitabine; CV: capecitabine and vinorelbine combination; GI: gastrointestinal. Bold: statistically significant difference

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
