# Peer review of "Oral Capecitabine-Vinorelbine Is Associated with Longer Overall Survival When Compared to Single-Agent Capecitabine in Patients with Hormone Receptor-Positive Advanced Breast Cancer"

_cancers, 2020, doi:10.3390/cancers12030617_

Round 1
Reviewer 1 Report
The study compares for the first time the effect of treatment with either capecitabine (C) or capecitabine-vinorelbine (CV) on PFS, OS and AE in patients with metastatic HER2-negative breast cancer. As well as confirming previous reports on the efficacy of C and CV treatment for mBC the study also shows that CV treatment has a positive effect on OS (but not PFS) in hormone receptor-positive patients, while also causing an overall increase in AE. The study proposes a potential mechanism of immune system modulation via analysis of NLR levels in patients treated with C or CV to explain why CV may improve OS while not improving PFS, however further study is needed to confirm this.
The number of patients used in the study is modest but not insignificant. The retrospective nature of the study is a limitation however the authors acknowledge this. The statistical analysis of the patient data is appropriate, has taken into consideration potential confounding factors and the conclusions are appropriately circumspect.
The overall impact of this study in its present form is limited in my opinion however the research conducted is sound.
Suggestions for improvement of the manuscript in it's present form:
- The Forest plots need to be improved. This may be a formatting issue however they are extremely hard to read. The font is very small and there appears to be "!" streaked across the plots, often obscuring the writing (particularly for Figure 2 A and C.
- Further comment/discussion on the impact of the effect of CV on AE compared to C. This analysis has been performed however the context or impact of this finding is somewhat glossed over.
- Although within the methods section the p<0.1 cutoff for univariate analysis should be outlined in the results section.
Reviewer 2 Report
In this study, the authors retrospectively analyzed a cohort of Her2- mBC patients treated with single C or combined CV for effectiveness. The manuscript is well-written, with appropriate description of methods.
There are multiple studies evaluating combination CV therapy in mBC patients. I think the study uncovers some new points, but should address some key concerns before publication.
1) Novelty: There are multiple studies on CV analysis in mBC patients. eg. http://ar.iiarjournals.org/content/29/2/667.full, https://www.sciencedirect.com/science/article/pii/S0923753419435965?via%3Dihub
Some studies are mentioned in the Discussion too. Some more text to discuss why this study is different from the other ones will highlight its value.
2) It looks like the patients getting CV tended to be younger. Do the authors know why? Line 79. Did the younger patients with CV have better ECOG PS?
3) Line 94. Patients with HR+ BC had better median PFS than TNBC with CV. Line 100: OS for HR+ is significant than TNBC for CV group as well.
- Is it because TNBC has poor prognosis to begin with (Fig S1A) or because with TNBC diagnosis, the patients had different Tax/Anthra/maintenance treatment?
- So can the authors analyze correlation with prior lines of chemo to see if that would correlate with response to C or CV?
- If PFS is not significant but OS is, does that mean patients are living longer with the disease instead of without? This can be discussed.
4) Line 147. The hypothesis about better immunomodulatory effects of CV in HR+ vs TNBC seems abrupt.
- What about the presence and absence of the actual ER or PR receptors and their correlation with the response or drug mechanism of action?
- Why did the authors choose NLR? It is a marker of immune response to many different types of stress and not just cancer.
- Line 218-219. does the prior chemo have any effect on the NLR and change the correlation between C and CV?
5) The authors should also analyze the pCR (complete response rates) or PR (partial response rate).
6) Is there any association with age, social features, smoking, etc? Is there a specific difference in response to C vs CV between ER+ and PR+ patients in the HR+ Her2- patients?
7) Authors should be careful to use "good safety profile" for the CV, given the increase in neutropenia and some AEs.
8) Minor formatting and English: Line 240. compared "to" TNBC, Line 352: in terms "of" PBS. The font is different from the rest of the text in Line 29 onwards, Line 234 onwards.
9) Lines 200-206: the meaning is unclear. What does "precocious" mean here? Maybe rewrite the sentences to clarify the meaning.
Reviewer 3 Report
Overall the manuscript cover a very interesting topic with high impact in patients management. The manuscript is well written and references are appropriately cited. below a couple of aspects that can be addressed before publication.
line 319. please provide a list of covariates considered independently associated and used in the model a, as weel as the covariates used in model b.
line 343 Authors stated that they set the statistical significance threshold at 0.05 for all analysis a part of the univariate analysis. Please, motivate this choice.
Reviewer 4 Report
In this study, authors have conducted a monocentric, retrospective analysis of 290 patients with HER2-negative metastatic breast cancer (mBC) treated with single-agent capecitabine (C) or capecitabine-vinorelbine (CV) to compare the progression-free survival (PFS), overall survival (OS) and incidence of adverse events. Authors have compared results between 127 patients treated with single-agent C and 163 patients treated with CV. Authors have shown that median PFS was similar in patients treated with single-agent C or CV, while CV was associated with significantly longer OS in patients with hormone receptor-positive (HR+) BC. Further, they have shown that CV was associated with higher incidence of adverse events when compared with single-agent C.
Overall, the manuscript presents some interesting data and conclude that while CV has advantage to improve the OS over single-agent C in HR+ mBC patients it is associated with significantly higher incidence of adverse effects.
There are some concerns in the manuscript which are provided below.
- Figure 2 and 3 showing forest plots are not very clear and readable. Authors need to replace these figures with better quality readable figures.
- Authors have not mentioned what was the follow-time for patients in this study? Please provide details.
- Authors have not mentioned anything about recurrence of breast cancer in the study population. It will be interesting to see what was the effect of C or CV on recurrence of breast cancer?
- Authors may also consider including ‘Quality of life’ or ‘Quality adjusted PFS’ to compare C vs. CV if data is available.
Round 2
Reviewer 4 Report
Authors have revised the manuscript thoroughly according to the comments and suggestions. The revised version version of manuscript could be accepted for publication.